# Growth and Welfare Status of Giant Freshwater Prawn (*Macrobrachium rosenbergii*) Post-Larvae Reared in Aquaponic Systems and Fed Diets including Enriched Black Soldier Fly (*Hermetia illucens*) Prepupae Meal

**DOI:** 10.3390/ani13040715

**Published:** 2023-02-17

**Authors:** Matteo Zarantoniello, Giulia Chemello, Stefano Ratti, Lina Fernanda Pulido-Rodríguez, Enrico Daniso, Lorenzo Freddi, Pietro Salinetti, Ancuta Nartea, Leonardo Bruni, Giuliana Parisi, Paola Riolo, Ike Olivotto

**Affiliations:** 1Department of Life and Environmental Sciences, Marche Polytechnic University, 60131 Ancona, Italy; 2Department of Agriculture, Food, Environment and Forestry, University of Florence, 50144 Firenze, Italy; 3Department of Agricultural, Food, Environmental and Animal Sciences, University of Udine, 33100 Udine, Italy; 4Mj Energy srl Società Agricola, Contrada SS. Crocifisso, 22, 62010 Treia, Italy; 5Department of Agricultural, Food and Environmental Sciences, Polytechnic University of Marche, 60131 Ancona, Italy

**Keywords:** giant freshwater prawn, sustainable aquaculture, hepatopancreas, circular economy, histology

## Abstract

**Simple Summary:**

The giant freshwater prawn culture is still challenged by dependence on diets containing high percentages of unsustainable and expensive marine-derived ingredients that are necessary to ensure the proper growth and welfare of this species. The present study proposed the replacement of fish meal and fish oil with black soldier fly full-fat prepupae meal enriched with spirulina in order to formulate two experimental diets intended for giant freshwater prawn, which were tested during a 60-day feeding trial in aquaponic systems. The present research represents an example of implementing more sustainable aquafeeds and farming techniques that promote both the valorization of circular economies and zero-waste concepts.

**Abstract:**

Due to the limited application of insect meal in giant freshwater prawn (*Macrobrachium rosenbergii*) culture, the present study aimed to (i) produce spirulina-enriched full-fat black soldier fly (*Hermetia illucens*) prepupae meal (HM) and (ii) test, for the first time, two experimental diets characterized by 3% or 20% of fish meal and fish oil replacement with full-fat HM (HM3 and HM20, respectively) on *M. rosenbergii* post-larvae during a 60-day feeding trial conducted in aquaponic systems. The experimental diets did not negatively affect survival rates or growth. The use of spirulina-enriched HM resulted in a progressive increase in α-tocopherol and carotenoids in HM3 and HM20 diets that possibly played a crucial role in preserving prawn muscle-quality traits. The massive presence of lipid droplets in R cells in all the experimental groups reflected a proper nutrient provision and evidenced the necessity to store energy for molting. The increased number of B cells in the HM3 and HM20 groups could be related to the different compositions of the lipid fraction among the experimental diets instead of a nutrient absorption impairment caused by chitin. Finally, the expression of the immune response and stress markers confirmed that the experimental diets did not affect the welfare status of *M. rosenbergii* post-larvae.

## 1. Introduction

The giant freshwater prawn *Macrobrachium rosenbergii* is one of the most cultured crustacean species worldwide due to its nutritional value, good meat quality, and high growth rate [1,2]. The production of *M. rosenbergii* is considered economical and more environmentally sustainable than that of other intensively reared species of crustaceans, as low stocking densities are often used [3,4]. However, this aquaculture sector is still challenged by the species’ dependence on diets containing high percentages of unsustainable and expensive marine-derived ingredients, such as fish meal (FM), which necessary to ensure proper growth and welfare [5,6].

In recent decades, great efforts have been made within *M. rosenbergii* farming to reduce the costs and the environmental footprint of dietary formulations by replacing FM with alternative protein sources of both plant and animal origin [7,8,9]. Plant-derived ingredients were first sought because of their wider availability and competitive costs [10]. In particular, it was first demonstrated that soybean meal could partially or totally replace FM in diets for *M. rosenbergii* without a negative effect on growth [11,12]. However, further studies on *M. rosenbergii* that coupled zootechnical performances with protein metabolism, antioxidant capacity, and intestinal microbiota demonstrated that FM replacement with soybean meal should not exceed 50% [9,13].

Monitoring the overall welfare of crustaceans through their physiological responses is of primary importance when new aquafeed ingredients are tested, especially for species such as the giant freshwater prawn that are frequently subject to infectious diseases with consequent economic losses[14,15]. In this regard, feed additives or probiotics that are able to enforce the resistance to pathological and environmental stresses of *M. rosenbergii* have been successfully used as a better alternative to chemicals and antibiotics [16,17,18]. Furthermore, to increase aquafeed sustainability, great attention has been given to alternative ingredients; for example, microbial biomass and insect meal, in addition to their immunostimulant features, may offer the possibility of replacing high amounts of FM as a major protein source [19,20]. In this context, promising results on both the growth performances and the immune responses of *M. rosenbergii* have been obtained by partially replacing FM with microbial biomass, such as yeasts [6] and microalgae [21,22,23], or by including insect meal in diets totally deprived of FM [24]. However, the high production costs and prices of microalgae and the fatty-acids (FA) profile of insects represent the main bottleneck that limits a wider use of those ingredients in aquaculture [25,26]. However, some insect species, such as the black soldier fly (*Hermetia illucens*), which have the proper protein tenor and amino acid profile and an appreciable content of bioactive molecules (i.e., lauric acid, chitin, and antimicrobial peptides) with antimicrobial and immunostimulatory features [27,28,29], are able to modulate their nutritional profile in relation to the growth substrate [30,31]. In this regard, the enrichment of the *H. illucens* growth substrate with microbial biomass represents an efficient method of exploiting the beneficial properties of both ingredients [32,33]. In particular, a recent study demonstrated that the 15% (*w*/*w*) inclusion of dried biomass from *Arthrospira platensis* (spirulina) in insects’ growth substrate resulted in a successful transfer of antioxidant molecules (tocopherols and carotenoids) of this cyanobacterium to *H. illucens* larvae, with consequent beneficial effects on fish-fed diets, including enriched insect meal [33].

*Arthrospira platensis* has gained notable attention for aquafeed formulation due to its protein content (55–70% dry weight), essential amino acids (methionine, cystine, and lysine), and pigments with antioxidant properties [34]. These pigments can efficiently preserve the muscle quality of fish, counteracting the oxidation of important molecules such as polyunsaturated fatty acids (PUFA) [35,36,37]. Spirulina has been directly used as an alternative ingredient to FM in diets for *M. rosenbergii*, with promising results on growth performance, body composition, and digestive and antioxidant activities [38,39]. Conversely, to the best of our knowledge, *H. illucens* prepupae meal (HM) has been scarcely tested in aquafeeds intended for this crustacean species [40].

The present study represents the first attempt to replace 3% and 20% of marine derivatives (both FM and oil) with enriched HM (15% *w*/*w* of *A. platensis* added to the growth substrate of *H. illucens* larvae) in diets intended for *M. rosenbergii* post-larvae. The 3% and 20% FM replacement levels were chosen, in accordance with a previous study [33], because they represent a feed supplement inclusion or an ecological inclusion level (a global 20% FM substitution in aquafeeds has a proper ecological impact), respectively. Particular emphasis was given to (i) growth performances and morphological indices, (ii) physical and chemical features of both muscle and exoskeleton, and (iii) the overall health status determined by the histological and molecular analyses on the hepatopancreas, which is the largest organ in the digestive tract of Decapoda and which is involved in absorption and metabolization of nutrients, as well as in the immune response [41]. To further improve prawn culture sustainability, a 60-day feeding trial was conducted in an aquaponic system, as different studies evidenced the suitability of this rearing method for *M. rosenbergii* culture [42,43].

## 2. Materials and Methods

### 2.1. Feeding Substrate Preparation and Insect Rearing

The feeding substrate of *H. illucens* larvae was mainly composed of coffee silverskin (a coffee industry by-product) prepared according to the method of Ratti et al. [33]. The coffee by-product was blended and reached a particle size of 0.4 ± 0.2 mm by using an Ariete 1769 food processor (De’ Longhi Appliances Srl, Treviso, Italy). The feeding substrate (coffee by-product) for *H. illucens* larvae was prepared by including 15% (*w*/*w*) of dried *Arthrospira platensis* biomass and by adding distilled water to reach final moisture of ~70% [44].

Six-day old *H. illucens* larvae (provided by Smart Bugs s.s., Ponzano Veneto, Treviso, Italy) were reared on the feeding substrate for 21 days in a climatic chamber in continuous darkness at 27 ± 1 °C, relative humidity of 65 ± 5%, and 0.3/cm^2^ density [45,46]. The feeding rate was 100 mg/day for each larva [47], maintained by adding, weekly, a new feeding substrate. The transition to the prepupal stage was identified by the change in tegument color from white to black [48]. Prepupae were then collected, washed, freeze-dried, and stored at −80 °C.

### 2.2. Prawns’ Diets Production

The freeze-dried full-fat *H. illucens* prepupae were ground to obtain fine powder for the prawns’ diet preparation (Retsch Centrifugal Grinding Mill ZM 1000, Retsch GmbH, Haan, Germany). Three test diets were formulated, according to the report of Goda [49], to be grossly iso-proteic (36% dry matter, DM), iso-lipidic (9% DM), and isoenergetic (18 MJ/kg DM), as far as possible. Ingredients and proximate composition of the experimental diets used in the present study are reported in Table 1. A control diet (named HM0) containing fish meal (FM), vegetable-derived ingredients, and fish oil (FO) was prepared and used as the basal formulation. The test diets that included black soldier fly prepupae meal (HM) were prepared by replacing, from the HM0 formulation, 3% (HM3) and 20% (HM20) of marine-derived ingredients (both FM and FO) with full-fat HM, previously obtained.

The test diets were manufactured in the pilot feed mill plant at the Department of Agrifood, Environment and Animal Science of the University of Udine (Udine, Italy). All the ingredients were weighed and placed in a mixer; during mixing, water was gradually added (approximately 400 g/kg) to obtain a wet consistency forming dough. Then, the dough was cold extruded with a meat mincer (provided with a knife at the die) to obtain 2 mm pellets that were subsequently dried at 40 °C for 48 h in a ventilated oven. The diets obtained were stored in sealed bags under vacuum and kept at −20 °C until use.

The test diets were analyzed for proximate composition following AOAC [50] procedures. The measure of moisture content was carried out by weight loss after drying samples in an oven at 105 °C until a constant weight was reached. The incineration to a constant weight, conducted in a muffle oven at 450 °C (HD 230, AMSE s.r.l.; Torino, Italy), was used to evaluate the ash content. Crude protein was determined as total nitrogen (N) through the Kjeldahl method and multiplying N by 6.25, while crude lipids were determined according to the Soxhlet ether method. Gross energy was measured by a calorimetric bomb (Adiabatic Calorimetric Bomb Parr 1261; PARR Instrument, IL, USA). For the HM proximate composition (% DM), please refer to Ratti et al. [33]. The ingredients, the proximate composition, the fatty acids profile, and the α-tocopherol and carotenoids content (determined as previously described in Ratti et al. [33]) of the test diets used in the present study are reported in Table 1.

### 2.3. Aquaponic Systems and Experimental Design

One thousand nine hundred seventy-one giant freshwater prawn post-larvae (initial weight: 0.10 ± 0.01 g) were provided by Miami Aqua-culture Inc, (Miami, FL, USA) and acclimated for two weeks in a single 500 L tank (mechanical, biological, and UV filtration provided by Panaque, Viterbo, Italy) at the aquaponics facility (MJ Energy srl, Treia, Macerata, Italy). Tank temperature was 28.0 ± 0.5 °C; ammonia and nitrite were <0.05 mg/L and nitrate <10 mg/L. At the end of the acclimation period, the prawns were randomly divided into nine Media Based Aquaponic Systems (720 L each of total volume) according to the 3 dietary treatments (three systems per experimental group, 219 specimens per tank). Each system consisted of a 1.56 m^2^ hydroponic unit for plant cultivation and a 600 L prawns’ tank.

Hydroponic unit. In each hydroponic unit, before the introduction of prawns in the aquaponic system, 15 lettuce (*Lactuca sativa*) seedlings, with an initial weight of 5.14 ± 0.47 g, were planted (density = 10 plants/m^2^). Expanded clay was added to each hydroponic unit to ensure both mechanical and biological filtration [51], and to guarantee physical support for plant growth. A 1900 L/h pump (Eheim GmbH & Co, Deizisau, Germany) was used to guarantee 3 water renewals per hour. From the prawns’ unit, water was pumped to the hydroponic one, then returned through a siphon equipped with synthetic foam to ensure an additional mechanical filtration.

Prawns’ unit. In the prawns’ tanks, a constant temperature (27.8 ± 0.2 °C) was maintained by TK500 chillers (Teco, Ravenna, Italy). The prawns were exposed to a natural photoperiod (14L/10D). Water samples were collected weekly for testing ammonia, nitrite, nitrate, and phosphate concentrations (Hanna reagents and HI83399 spectrophotometer; Hanna instruments, Villafranca Padovana, Italy).

The feeding trial duration was 60 days, during which the prawns almost tripled their weight. The prawns were hand-fed the experimental diets, provided at a feeding rate of 10% body weight, according to Posadas [52] (divided into two equal amounts, one in the morning and one in the afternoon), as follows: (i) prawns fed the control diet (the HM0 group), and (ii) prawns fed diets characterized by 3% or 20% of FM and FO replacement with HM (the HM3 and HM20 groups, respectively). The daily feeding ration was computed every 2 weeks by weighing a representative number of specimens per tank The feed particle size was 0.3 mm (obtained by mincing and sieving the pellets extruded during the preparation of the test diets). At the end of the trial, the prawns were sacrificed by thermal shock for individual measurements and for the excision of the cephalo-thorax for collection of hepatopancreas samples (which were properly stored for further analyses).

### 2.4. Survival Rate, Growth Performances, and Morphological Features

At the end of the trial, all the prawns were counted and weighted to calculate survival rates (SRs) and growth performances (weight gain, WG; relative growth rate, RGR; specific growth rate, SGR), using the following formulae:SR (%) = (final number of prawns/initial number of prawns) × 100(1)
WG (g) = FBW (final body weight, g) − IBW (initial body weight, g)(2)
RGR (%) = (WG/IBW) × 100(3)
SGR (% day^−1^) = [(ln FBW − ln IBW)/days of trial] × 100(4)

For the morphological characterization, a different set of 100 prawns from each tank was randomly collected to (i) measure FBW, total length (TL), and muscle lengths (ML), and (ii) calculate the incidence of both muscle weight (MW) and the weight of head and exoskeleton (HEW) on the final body weight (FBW), as well as the condition factor (K), as follows:Incidence of MW = MW/FBW × 100(5)
Incidence of HEW = HEW/FBW × 100(6)
K = FBW/TL^3^(7)

### 2.5. Histological Analysis

At the end of the trial, hepatopancreas samples were obtained from five prawns per tank (15 prawns per dietary group) and immediately fixed in Bouin’s solution for 24 h at 4 °C, then washed with ethanol (70%) [53]. The samples were dehydrated in 80%, 95%, and 100% ethanol solutions, washed with xylene, and embedded in paraffin (Bio-Optica, Milano, Italy). The solidified paraffin blocks were cut with a Leica RM2125 RTS microtome (Leica, Nussloch, Germany) to obtain 5 µm sections that were subsequently stained with Mayer hematoxylin and eosin Y (Merck KGaA, Darmstadt, Germany). Stained sections were examined (×20 and ×40 magnification) using a Zeiss Axio Imager.A2 (Zeiss, Oberkochen, Germany) microscope equipped with a combined color digital camera (Axiocam 105, Zeiss, Oberkochen, Germany), and image analysis was performed using the ZEN 2.3 software (Zeiss, Oberkochen, Germany). Histological analyses were focused on the middle portion of each hepatopancreas tubule (called B cell zone), as this part includes the mature stage of B and R cells [41]. The relative abundance of B and R cells, as well as the abundance of lipid droplets in R cells (an indication of the degree of lipid accumulation), the tubule diameter, and the height of the epithelium, were measured on 20 randomly selected tubules per section (3 sections per prawn, 15 prawns per dietary treatment). The score-assignment criteria for B and R cells’ relative abundance and R cells’ vacuolization are reported in Table 2.

### 2.6. RNA Extraction and cDNA Synthesis

Total RNA was extracted from hepatopancreas samples of 5 prawns from each tank (15 prawns per dietary group) using RNAzol RT reagent (Merck KGaA, Darmstadt, Germany) and finally eluted in RNase-free water (Qiagen, Hilden, Germany; 20 µL) and stored at −80 °C [54]. The NanoPhotometer P-Class (Implen, München, Germany) was used to determine RNA concentration, while RNA integrity was checked by GelRed^TM^ staining of 28S and 18S ribosomal RNA bands on 1% agarose gel. The iScript cDNA Synthesis Kit (Bio-Rad, Hercules, CA, USA) was used to synthesize the cDNA from 1 μg of RNA.

### 2.7. Real-Time PCR

PCRs were performed in an iQ5 iCycler thermal cycler (Bio-Rad, Milano, Italy) setting a 96-well plate. For each sample, reactions were set mixing 1 μL of 1:10 diluted-cDNA, 5 μL of fluorescent intercalating agent (2× concentrated iQ ™ Sybr Green, Bio-Rad, Milano, Italy) and 0.3 μM of both forward and reverse primer [55]. For all reactions, the thermal profile was 3 min at 95 °C, then 45 cycles of 20 s at 95 °C, 20 s at the annealing temperature, specific for each primer and reported in Table 3, and 20 s at 72 °C. In all cases, one single peak was detected in the melting curve analyses, no amplification products were observed in negative controls, and no primer-dimer formations were observed in control templates. The relative quantification of the expression of genes involved in molting regulation (juvenile hormone epoxide hydrolase, *jheh*), enzymatic hydrolysis of chitin (chitinase 3, *chit3*), protein digestion (cathepsin L, *catL*), stress response (heat shock protein 90, *hsp90*), and immune response (α2-macroglobulin, *α2m*) was performed. Beta-actin (*β-actin*) and 18S ribosomal protein (*18s*) were used as housekeeping genes. Amplification products were sequenced, and homology was verified. The iQ5 optical system software version 2.0 (Bio-Rad), including GeneEx Macro iQ5 Conversion and GeneEx Macro iQ5 files, was used to analyze PCR data.

### 2.8. Physical and Chemical Analyses

For each dietary group, 4 pools consisting of the homogenized muscle of 12 prawns were prepared. pH was measured with a pH-meter METTER TOLEDO (Metter-Toledo, Schwerzenbach, Switzerland). Moisture was calculated gravimetrically by freeze-drying each pool. Starting from the freeze-dried material, total lipids and fatty acid (FA) profiles were assessed, following Folch et al. [58]. The primary oxidative products (conjugated dienes, CD), as well as the secondary lipid oxidation products (thiobarbituric acid reactive substances, TBARS), were analyzed according to Srinivasan et al. [59] and Vyncke [60], respectively. Results were expressed as mmol hydroperoxides (mmol Hp) and malondialdehyde equivalents (MDA-eq.) on 100 g freeze-dried prawn muscle for CD and TBARS, respectively.

For each dietary group, 4 pools consisting of the homogenized exoskeleton of 12 prawns were prepared to analyze the chitin content, following Hahn et al. [61].

### 2.9. Statistical Analysis

The aquaponic systems were used as the experimental unit for data related to survival rates and growth performance, while individual specimens were the experimental units for all the remaining analyses. All data were checked for normality (Shapiro–Wilk test) and homoscedasticity (Levene’s test). Data for survival rates, growth parameters, histological analysis, and relative quantification of gene expression were analyzed through one-way analysis of variance (ANOVA) followed by Tukey’s multiple comparison post hoc test, performed using the software package Prism 8 (GraphPad software version 8.0.2, San Diego, CA, USA). Significance was set at *p* < 0.05.

Data on morphological traits and on physical and chemical characteristics were assessed with a one-way ANOVA followed by a Tukey’s test using the free software environment R [62], with significance set at *p* < 0.05. All data were expressed as mean ± standard deviation (SD).

## 3. Results

### 3.1. Water Quality Parameters

No significant differences were observed among the experimental groups in terms of nitrate concentration (53.7 ± 1.3, 52.9 ± 1.1, and 52.8 ± 1.2 mg/L for HM0, HM3, and HM20, respectively) and phosphate concentration (1.5 ± 0.2, 1.7 ± 0.1, and 1.7 ± 0.2 mg/L for HM0, HM3, and HM20, respectively). Ammonia and nitrite concentrations were lower than 0.05 mg/L at each weekly measurement in all the experimental groups.

### 3.2. Survival Rate, Growth Performance and Morphological Features

All the daily-provided feed was consumed by the prawns of each tank within 15 min. Survival rates, growth performance, and the morphological features of *M. rosenbergii* are reported in Table 4. No significant differences were evident among the experimental groups considering the indices calculated, except for HEW, for which prawns from the HM20 group exhibited a significantly higher value than those from both the HM0 and HM3 groups (*p* < 0.001).

### 3.3. Histological Analysis

No histopathological alterations were evident in all the tubules analyzed. Hepatopancreas samples from all the experimental groups showed a well-organized tubular structure (Figure 1 and Figure 2) with diffused R cells characterized by highly abundant lipid droplets (Table 5). However, higher number of B cells was detected in hepatopancreas samples from prawns fed the HM3 and HM20 diets, compared with those fed on the HM0 diet. No significant differences were evident among the experimental groups in terms of tubule diameter or epithelium height (Table 5).

### 3.4. Gene Expression

As evidenced in Figure 3, no significant differences were evident among the experimental groups in terms of *jheh*, *chit3*, *hsp90*, or *α2m* gene expression. However, *catL* was significantly upregulated in the HM3 and HM20 groups, compared with the HM0 group (Figure 3c).

### 3.5. Physical and Chemical Analyses

No significant differences were detected among the experimental groups considering pH, moisture muscle samples, and chitin content of exoskeleton samples from prawns fed the different experimental diets (Table 6).

The total lipid content and FA profile of muscle samples from prawns fed the different experimental diets are shown in Table 7. No significant differences were detected among the experimental groups in terms of total lipid, SFA, MUFA, and n3 PUFA contents. However, the n6 PUFA content was significantly higher in the HM3 and HM20 groups compared to the HM0 group (*p* < 0.001). Considering the FA profile, palmitic acid (C16:0) was the most represented SFA in all the experimental groups and its content was significantly higher in the HM0 group compared to the other groups (*p* < 0.001). Oleic acid (C18:1n9) was the most abundant MUFA in all the experimental groups, with significantly higher values in the HM0 and HM3 groups compared to the HM20 group (*p* < 0.001). Finally, considering PUFA, linoleic acid (C18:2n6; LA) and arachidonic acid (C20:4n6; ARA) contents were significantly higher in the HM3 group than in the other groups (*p* < 0.001). EPA (C20:5n3) content did not show significant differences among the experimental groups, while DHA (C22:6n3) content highlighted a progressive significant reduction from the HM0 group to the HM20 group (*p* = 0.00125).

As shown in Table 8, CD, i.e., primary oxidative products, did not significantly vary among the experimental groups. However, TBARS, i.e., the secondary oxidative products, were significantly higher in the HM0 group than in the HM20 group (*p* = 0.0321), while the HM3 group showed intermediate values.

## 4. Discussion

The use of HM as an alternative and more sustainable ingredient in aquafeed formulation has been widely studied in fish [63], but less in crustaceans. Several studies have been performed on the Pacific white shrimp (*Litopenaeus vannamei*) using different dietary inclusion levels of *H. illucens* larvae meal, obtaining promising results in terms of growth performances, antioxidant and immune response, digestive enzyme activity, intestinal microbiota, histomorphology of hepatopancreas, and muscle biochemical composition [64,65,66]. However, to the best of our knowledge, *H. illucens* has been scarcely explored as a possible alternative to FM in dietary formulations intended for a widely farmed Decapoda, such as the giant freshwater prawn [40]. The present study represents the first attempt to evaluate the replacement of FM and FO with spirulina-enriched full-fat HM on *M. rosenbergii* post-larvae reared in aquaponic systems to further improve aquaculture sustainability, promoting a zero-waste concept [67]. The three experimental diets tested in the present study, using aquaponic systems, resulted in comparable survival and growth rates, confirming the suitability of the diets and the farming technique for *M. rosenbergii* [42,68]. HM represents a good source of proteins and it was successfully used in the *M. rosenbergii* diet, which is known to require a relatively high dietary protein content (37% on DM) to support growth performances [69,70]. Accordingly, growth performances and morphological features were not negatively affected by the HM dietary inclusions, which additionally led to a higher HEW in prawns from the HM20 group. These results are in line with previous studies that demonstrated that a replacement of FM with *H. illucens* larvae meal up to 20% did not affect the growth performances of the Pacific white shrimp [64,65,66,71,72]. However, it should be pointed out that the use of high-protein diets in crustacean aquaculture generally causes a consequently high nitrogenous waste production that can interfere with water quality [73]. In this sense, as demonstrated by water quality analyses, aquaponic systems represented a valid method of exploiting this metabolic waste as a macronutrient for plant growth [3,42], avoiding economic losses due to prawn mortality linked to high ammonia levels and possible environmental issues [74,75].

The addition of spirulina in the insects’ rearing substrate resulted in an increase in α-tocopherol from the HM3 to HM20 diets and in the presence of carotenoids in the HM20 diet, respectively, highlighting that this enrichment procedure can be a valid method to transfer these important antioxidant molecules from spirulina to *H. illucens* and, then, to the experimental diets. The provision of diets enriched with α-tocopherol and carotenoids has possibly reduced the secondary oxidative products in muscle samples from the HM0 group to the HM20 group, confirming the role of these molecules in preserving the muscle-quality traits [35,36,37,76].

However, the inclusion of full-fat HM in aquafeeds is usually associated with a dietary fatty acid profile alteration, especially regarding the SFA and the n3 PUFA contents [32,77]. Accordingly, the experimental diets tested here showed a slight increase in SFA content and a parallel wider decrease in n3 PUFA (especially in terms of EPA and DHA) from HM0 to HM20. Comparable amounts of n6 PUFA (largely constituted by LNA) were observed among the experimental diets. It has been demonstrated that the balance (more than the relative content) of dietary n3 and n6 PUFA is a crucial factor for the fatty acid requirement in prawns’ nutrition [78]. In fact, despite the necessity to provide n3 PUFA [79], freshwater prawns such as *M. rosenbergii* might achieve the desired growth performance by relying on the dietary n6 PUFA [69]. This is a consequence of the terrestrial origin of the giant freshwater prawn’s natural diet, rich in fatty acids from the n6 series [80]. The absence of a preference for either n3 or n6 series fatty acids to fulfil the PUFA requirements [81] sustained a proper growth of each experimental group. In addition, the tissue fatty acid profile of giant freshwater prawn generally reflects the dietary profile, due to the scarce ability of this crustacean to synthesize long-chain PUFA from shorter-chain precursors [82,83]. Accordingly, in the present study, the proportion among the fatty-acid classes in the diets reflected that of the muscle samples. The significantly higher amount of DHA in the muscle samples of prawns from the HM0 group emphasizes the importance of providing diets characterized by a proper amount of long-chain PUFA, in order to guarantee the quality of the final product and to meet the consumer’s requirements. However, while the experimental diets showed a higher DHA content compared to EPA, an opposite result was obtained in muscle samples, reflecting the ability of *M. rosenbergii* to synthesize EPA from DHA and its bioconversion inability in the opposite direction [83].

The dietary lipid content can also affect the lipid deposition in hepatopancreas; in particular, the R cells are known to be the main site for lipid storage and are thus considered to be a crucial indicator of crustaceans’ nutritional condition [84,85]. R cells are the most abundant cell type in the hepatopancreas, but nutritional stress can lead to a decrease in their prevalence in the tubules and vacuolation, with a consequent epithelium atrophy [84,86]. Accordingly, Wang et al. [87] evidenced vacuolar degeneration coupled with an alteration of the epithelium structure in Pacific white shrimp when the replacement of FM with *H. illucens* larvae meal exceeded 60%. However, in the present study, which used lower replacement percentages, epithelium height and lipid storage were not affected by the provision of HM diets, as demonstrated by the comparable frequency of R cells and the high abundance of lipid droplets in all the experimental groups.

Dietary nutrients are absorbed and stored by the R cells to be subsequently mobilized to sustain energy-demanding processes such as molting [88]. The massive presence of lipid droplets in R cells in all the experimental groups reflected a proper nutrient provision by the experimental diets and evidenced the necessity of prawns to store energy reserves during the inter-molt stage [83]. Crustaceans periodically degrade the old exoskeleton in favor of a newly synthesized one (pre-molt phase) and, during molting, they face a rapid uptake of water that increases the animal size. After a post-molt phase, in which the hardness of the new exoskeleton is improved, the water absorbed during molting is gradually replaced by tissue growth and reserves are stored (inter-molt phase) [83]. This process is under hormonal control. In particular, methyl farnesoate is the crustacean analogue of the pleiotropic juvenile hormone (JH) that is typical of insects and, like JH, it is involved in several processes including molting [89]. This hormone, together with the ecdysone, gradually increases in the pre-molt phase, reaching the maximum levels near molting [90,91]. However, a key determinant for successful molting is the rapid decrease in methyl farnesoate due to its JHEH-mediated inactivation. It has been demonstrated that in *M. rosenbergii*, the higher *jheh* expression in the pre-molt stage led to a successful molt in a short time [92]. The crustacean hepatopancreas is considered the main site for methyl farnesoate inactivation due to the highest level of *jheh* expression [92]. In the present study, the absence of significant differences in *jheh* gene expression represented further confirmation that the molting cycle (and thus somatic growth) and its regulation were not affected by the different dietary treatments.

The activity of digestive enzymes can have a direct effect in supporting molting and growth because of the breakage of the dietary high-weight biomolecules into more available forms required for metabolism or storage [38]. In this regard, the primary role of the hepatopancreas is the digestive enzymes’ synthesis and secretion, the digestion of the ingested food, and the subsequent nutrients’ uptake [93]. In particular, B cells produce digestive enzymes that are involved in intracellular digestion and concentrate in the vacuole the absorbed materials, and secrete them in the tubular lumen at the end of the digestive process for reabsorption by R cells [88]. In the present study, a higher relative abundance of B cells was detected in the HM3 and HM20 groups than in the HM0 group. This result was fully supported by the expression of *catL* which codifies for the cathepsin L, a protease stored in the digestive vacuole of B cells and secreted into the tubular lumen of the hepatopancreas for extracellular digestion [94]. In this regard, it should be pointed out that Hu et al. [94] detected *catL* mRNA only in F cells but not in mature B cells that only host the active enzyme. However, several authors suggested that F cells are the precursors of B cells, and from an F cell to a B cell there is a continuum of transition (referred to as F/B cells) [95]. Therefore, although we only measured the mRNA relative abundance of *catL*, this result supports the abundance of B cells due to the fast synthesis of cathepsin L and the quick transition from F cell type to B cell type [94].

An increased B cells population in the hepatopancreas is usually associated with the reduced bioavailability of dietary nutrients [96,97]. Several studies on fish reported that one of the possible drawbacks related to the dietary inclusion of HM can be attributed to chitin, which, due to its complex matrix, can impair nutrient absorption, creating a barrier on the absorptive epithelium [98,99,100]. In addition, Kumar et al. [101] found that *M. rosenbergii* that were fed post-larvae with 10% purified chitin had a lower growth rate than those fed with 5% purified chitin, confirming that in crustaceans this complex molecule can act as a barrier against nutrient absorption. However, crustaceans possess specific chitinases expressed in the hepatopancreas, among which is the CHIT3 that is involved in the degradation of chitin-containing feeds; chitinases are part of the natural diet of this species [83,102]. In the present study, the *chit3* gene expression in the hepatopancreas did not significantly vary among the experimental groups, highlighting that HM diets did not enhance the production of this chitinase. For that reason, instead of a nutrient absorption impairment caused by chitin, the increased B cells number observed in the HM3 and HM20 groups could be related to the different compositions of the lipid fraction among the experimental diets. In fact, B cells may increase in number and size in response to high dietary lipids, producing and recycling fat emulsifiers to counteract their hydrophobic features that slow down the digestive processes and to facilitate their endocytosis [103,104]. As the diets were iso-lipidic, this conclusion could possibly be related to the dietary amount of SFA and MUFA that increased with the increasing dietary inclusion levels of HM and to the ability of *M. rosenbergii* to digest and use these fatty-acid classes as an energy source [80,83].

Finally, the hepatopancreas, in addition to being a sensitive indicator for the nutritional status, is used to monitor the health conditions of crustaceans, as it is involved in the immune and stress response [96,105,106]. Crustaceans only possess innate immunity due to the absence of cells from the myeloid lineage that are able to produce specific antibodies to face repetitive infections [107]. Specifically, crustaceans have a rapid innate immunity consisting of various prophenoloxidase (proPO) system immune-related proteins, including α2-macroglobulin, which acts against pathogens and regulates the proPO activation [108]. In addition, it has been demonstrated that the chitin-related genes expressed in the hepatopancreas of crustaceans may be involved in the immune response and disease resistance through direct or indirect regulation [109,110]. The expression of the immune response marker *a2m*, supported by the absence of significant differences in terms of *chit3* gene expression, confirmed that the experimental diets used in the present study did not affect the health status of *M. rosenbergii*. This conclusion is further supported by the expression of *hsp90*, a stress-related biomarker that can also play a significant role in host immunity and the health of crustaceans [111].

## 5. Conclusions

The experimental diets used in the present study properly sustained survival and growth and fulfilled the PUFA requirement of giant freshwater prawn post-larvae. The dietary lipid content of each experimental diet led to a massive presence of lipid droplets in R cells in all the experimental groups, allowing a proper storage of energy sources to sustain molting processes. However, the diverse composition of the lipid fraction among the experimental diets resulted in an increase in B cells abundance in the HM3 and HM20 groups. In addition, the use of spirulina-enriched HM in the diets allowed the transfer of important bioactive molecules to the prawns, which possibly played a crucial role in preserving muscle-quality traits. Finally, the hepatopancreas health status was preserved in all the experimental groups, as evidenced by the histological analyses and the expression of immune and stress markers.

The present study represents an example of how *M. rosenbergii* culture can be implemented with more sustainable aquafeeds and farming techniques such as aquaponic systems, to promote both the valorization of the circular economy and the zero-waste concepts in aquaculture.

## Figures and Tables

**Figure 1 animals-13-00715-f001:**
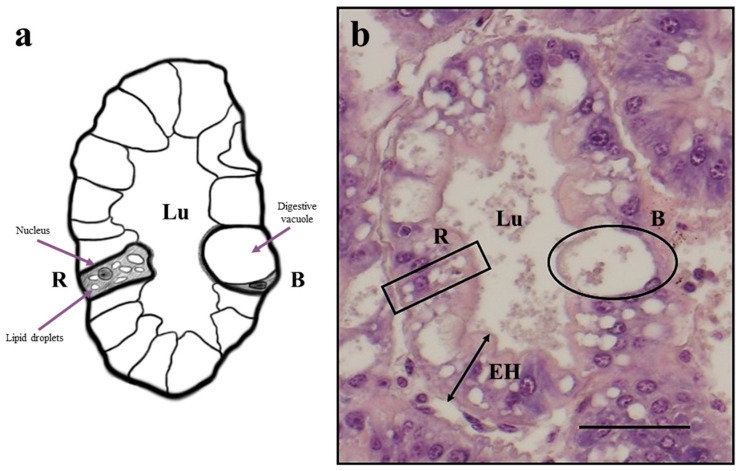
(**a**) Schematic representation of transversal section of hepatopancreas tubule from highlighting a detailed example of R cell and B cell (author: Matteo Zarantoniello). (**b**) Example of histomorphology of transversal section of a hepatopancreas tubule (middle portion) from HM0 group. Scale bar: 50 µm. Abbreviations: R, R cell; B, B cell; EH, epithelium height; Lu, lumen.

**Figure 2 animals-13-00715-f002:**
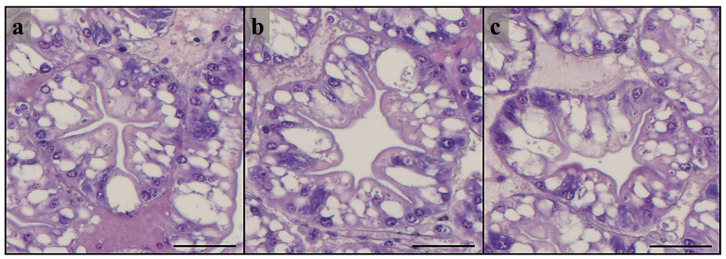
Histomorphology of transversal sections of hepatopancreas tubules (middle portion) of giant freshwater prawn fed post-larvae over 60 days with the test diets. (**a**) HM0, (**b**) HM3, and (**c**) HM20. Scale bars: 50 µm.

**Figure 3 animals-13-00715-f003:**
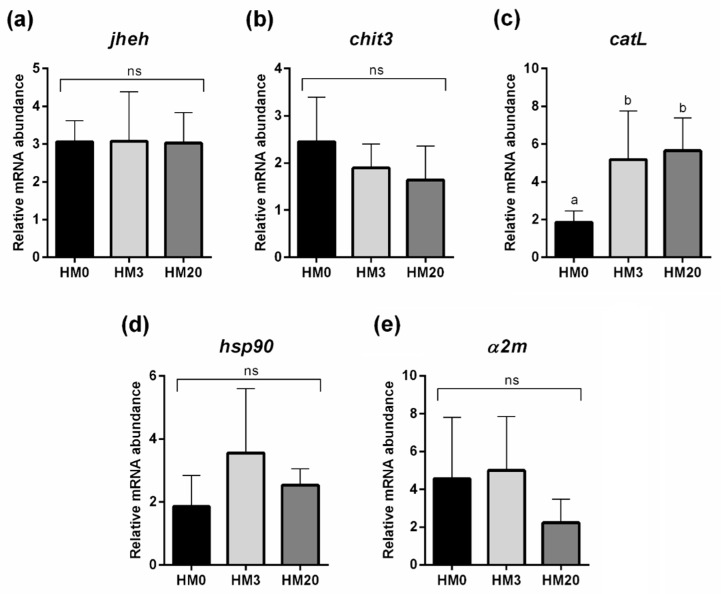
Relative mRNA abundance of genes analyzed in hepatopancreas samples of giant freshwater prawn fed post-larvae over 60 days with the experimental diets (HM0, HM3, and HM20). (**a**) *jheh*, (**b**) *chit3*, (**c**) *catL*, (**d**) *hsp90*, and (**e**) *α2m*. Values are shown as mean ± standard deviation (*n* = 15). ^a,b^ Different letters indicate significant differences among experimental groups (*p* < 0.05). Abbreviations: ns, no significant differences.

**Table 1 animals-13-00715-t001:** Ingredients (g/kg as a fed-basis), proximate composition (% on dry matter, DM), fatty acids profile, and α-tocopherol and carotenoid content of the test diets.

	HM0	HM3	HM20
**Ingredients**			
Fish meal	400	388	317
Soy protein concentrate	60	60	60
*Hermetia illucens* meal	-	13	88
Wheat bran	190	190	190
Wheat meal	150	150	150
Rice bran	115	115	115
Fish oil	40	39	35
Soya lecithin	10	10	10
Mineral and vitamin premix ^#^	15	15	15
Binder	20	20	10
**Proximate composition (% DM)**			
Dry matter	92.3	93.8	92.5
Crude protein	36.1	36.0	35.8
Crude lipid	9.2	9.7	9.6
Ash	14.2	13.4	13.2
Gross energy (MJ/kg)	18.8	18.9	18.9
**FA profile (g/100g of total FAME)**			
C12:0	0.16	0.80	4.67
C14:0	5.03	5.34	5.33
C16:0	18.97	20.78	19.49
C16:1n7	5.60	5.63	6.02
C18:0	3.98	4.36	3.86
C18:1n9	14.09	14.11	14.00
C18:1n7	2.38	2.38	2.29
C18:2n6, LNA	11.15	11.23	11.72
C18:3n3, ALA	1.58	1.50	1.42
C18:4n3	1.81	1.63	1.29
C20:1n9	2.79	2.71	3.30
C20:5n3, EPA	7.44	6.53	5.21
C22:1n11	3.68	3.50	4.69
C22:6n3, DHA	12.67	11.14	8.23
*ΣSFA*	30.02	33.46	36.15
*ΣMUFA*	29.72	29.50	31.60
*Σn6 PUFA*	12.78	12.67	13.07
*Σn3 PUFA*	25.15	22.23	17.30
*Σn4 PUFA*	1.39	1.39	1.26
**α-tocopherol and carotenoids content (mg/kg)**			
α-tocopherol	<LOD	2.22	12.24
β-carotene	<LOD	<LOD	6.58
zeaxanthin	<LOD	<LOD	2.48

^#^ Mineral and vitamin premix as reported by Goda [49]. Values are reported as mean of triplicate values. FAME: fatty acid methyl ester; LNA, ALA, EPA, and DHA: linoleic, α-linolenic, eicosapentaenoic, and docosahexaenoic acids, respectively; SFA, MUFA, and PUFA: saturated, monounsaturated, and polyunsaturated fatty acids, respectively; LOD: limit of detection. The fatty acids C13:0, C14:1n5, iso-C15:0, anteiso-C15:0, C15:0, iso-C16:0, C16:1n9, C16:2n4, C17:0, C16:3n4, C17:1, C16:4-n1, C18:2n4, C18:3n6, C18:3n4, C18:4n1, C20:0, C20:1n11, C20:1n7, C20:2n6, C20:3n6, C20:4-n6, C20:3-n3, C20:4-n3, C22:0, C22:1-n9, C22:1-n7, C22:2-n6, C21:5-n3. C22:4-n6, C22:5-n6, C22:5-n3, and C24:0, quantified at levels <1 g/100 g total fatty acid methyl esters, were utilized for calculating the fatty acid classes but they are not reported in the table.

**Table 2 animals-13-00715-t002:** Score-assignment criteria for B and R cells’ relative abundance and R cells’ lipid droplets abundance, measured in histological sections from the middle portion of hepatopancreas tubules.

Parameter	Score	Description
	+	Scarce
B cells’ relative abundance	++	Diffuse
	+++	Highly abundant
	+	Scarce
R cells’ relative abundance	++	Diffuse
	+++	Highly abundant
	+	Scarce
R cells’ lipid droplets abundance	++	Diffuse
	+++	Highly abundant

**Table 3 animals-13-00715-t003:** Sequences, identification numbers, and annealing temperature (AT) of primers used in the present study.

Gene	Forward Primer (5′-3′)	Reverse Primer (5′-3′)	AT (°C)	ID Number	References
*jheh*	GCCATTGTTGACGAAGCC	GCCACCCTGGAGGTAGAA	58	KM886343	[56]
*chit3*	GGGCTTGGCTGGTTGTAT	GGTGGAGGTGGAGTTGGA	58	LT574899	[56]
*catL*	CCTCTTGGTCGTCGCCTTAG	CCTCTTGGTCGTCGCCTTAG	59	KY474042	[56]
*hsp90*	GAAGGAAAGGGACAAGGA	GGTCCATAAAGGCTTGGT	58	GU319963.1	[57]
*α2m*	CTCGGCCATCTTATCCGTATG	GGGAGCGAAGTTGAGCATGT	58	ABK60046	[24]
*18s*	TCCGTAAGGACCTGTATGCC	TCGGGAGGTGCGATGATTTT	57	AY651918.2	[16]
*β-actin*	CTGTTACGGGTGACGGAGAA	TCGGAAGAGTCCCGCATT	57	GQ131934	[56]

**Table 4 animals-13-00715-t004:** Survival rate, growth performance, and morphological features of giant freshwater prawn fed the experimental diets post-larvae over the 60-day trial.

	HM0	HM3	HM20	*p* Value
SR (%)	82.8 ± 0.9	82.2 ± 0.9	81.3 ± 1.1	0.242
IBW (g/prawn)	0.10 ± 0.01	0.10 ± 0.02	0.10 ± 0.02	0.368
FBW (g/prawn)	0.34 ± 0.03	0.35 ± 0.03	0.33 ± 0.01	0.606
RGR (%)	245.0 ± 12.1	253.7 ± 14.51	235.4 ± 5.78	0.226
SGR (% day^−1^)	2.53 ± 0.10	2.69 ± 0.09	2.52 ± 0.03	0.031
MW (%)	39.5 ± 2.4	39.6 ± 3.1	40.3 ± 3.0	0.053
HEW (%)	53.0 ± 4.0 ^a^	52.1 ± 3.5 ^b^	55.5 ± 3.8 ^b^	<0.001
K	1.29 ± 0.2	1.27 ± 0.2	1.26 ± 0.2	0.701

Values are shown as mean ± SD (*n* = 3 for survival rate and growth performance; and *n* = 100 for morphological features). ^a,b^ Different letters indicate significant differences among experimental groups (*p* < 0.05). Abbreviations: SR, survival rate; IBW, initial weight; FBW, final weight; WG, weight gain; RGR, relative growth rate; SGR, specific growth rate; MW, incidence of muscle weight on FBW; HEW, incidence of the weight of head and exoskeleton on FBW.

**Table 5 animals-13-00715-t005:** Histological indices and morphometric evaluation of tubule diameter and epithelium height measured in hepatopancreas tubules (middle portion) of giant freshwater prawn fed the experimental diets over the 60-day trial.

	HM0	HM3	HM20
B cells’ relative abundance	++	+++	+++
R cells’ relative abundance	++	++	++
R cells’ lipid droplets abundance	+++	+++	+++
Tubule diameter	135.6 ± 6.1	134.6 ± 5.2	136.4 ± 6.3
Epithelium height	63.9 ± 3.1	64.2 ± 2.1	67.1 ± 2.2

Values of tubule diameter and epithelium height are shown as mean ± SD (*n* = 15).

**Table 6 animals-13-00715-t006:** pH and moisture (g/100g of muscle) measured in the muscle and chitin content (g/100g of freeze-dried exoskeleton) measured in the exoskeleton from giant freshwater prawns fed the experimental diets over the 60-day trial.

	HM0	HM3	HM20	*p* Value
pH	6.72 ± 0.07	6.63 ± 0.07	6.60 ± 0.05	0.189
Moisture	76.9 ± 0.4	76.7 ± 1.0	77.2 ± 0.6	0.550
Chitin	12.2 ± 0.7	12.2 ± 0.7	13.3 ± 0.5	0.056

Values are shown as mean ± SD (*n* = 4).

**Table 7 animals-13-00715-t007:** Total lipid content (g/100 g of freeze-dried muscle) and fatty acid profile (g/100 g of total FAME) of giant freshwater prawn fed the experimental diets.

	HM0	HM3	HM20	*p* Value
Total lipids	5.41 ± 0.06	4.87 ± 0.32	5.19 ± 0.33	0.050
Fatty acids				
C14:0	1.63 ± 0.06	1.58 ± 0.03	1.66 ± 0.09	0.258
C16:0	23.4 ± 0.39 ^b^	22.56 ± 0.18 ^a^	22.14 ± 0.21 ^a^	<0.001
C16:1n7	2.07 ± 0.04 ^a^	2.32 ± 0.12 ^b^	2.22 ± 0.10 ^ab^	0.0141
C18:0	9.13 ± 0.11 ^a^	9.04 ± 0.08 ^a^	9.48 ± 0.10 ^b^	<0.001
C18:1n9	15.60 ± 0.14 ^b^	15.34 ± 0.21 ^b^	14.80 ± 0.26 ^a^	<0.001
C18:1n7	3.72 ± 0.05 ^a^	3.93 ± 0.04 ^b^	4.03 ± 0.09 ^b^	<0.001
C18:2n6, LNA	8.47 ± 0.17 ^a^	8.20 ± 0.17 ^a^	9.02 ± 0.12 ^b^	<0.001
C18:3n3, ALA	0.77 ± 0.01 ^a^	1.03 ± 0.02 ^c^	0.82 ± 0.01 ^b^	<0.001
C20:1n9	1.02 ± 0.03 ^b^	0.88 ± 0.01 ^a^	1.19 ± 0.04 ^c^	<0.001
C20:4n6, ARA	3.65 ± 0.10 ^a^	4.64 ± 0.25 ^b^	3.89 ± 0.19 ^a^	<0.001
C20:5n3, EPA	16.63 ± 0.33	16.49 ± 0.06	16.99 ± 0.47	0.141
C22:6n3, DHA	7.89 ± 0.20 ^b^	7.55 ± 0.08 ^ab^	7.19 ± 0.22 ^a^	0.00125
*ΣSFA*	37.10 ± 0.54	36.42 ± 0.22	36.57 ± 0.29	0.0694
*ΣMUFA*	23.23 ± 0.25	23.38 ± 0.14	23.19 ± 0.40	0.61
*Σn3 PUFA*	26.15 ± 0.50	25.82 ± 0.24	25.81 ± 0.73	0.613
*Σn6 PUFA*	13.08 ± 0.10 ^a^	13.95 ± 0.13 ^b^	14.03 ± 0.15 ^b^	<0.001

Values are shown as mean ± SD (*n* = 4). ^a–c^ Different letters indicate statistically significant differences among the experimental groups (*p* < 0.05). The fatty acids C12:0, C15:0, C16:0, C16:1n9, C17:0, C16:2n4, C16:3n4, C17:1, C16:4n1, C18:2n4, C18:3n6, C18:3n4, C18:4n3, C20:0, C20:1n11, C20:1n7, C20:2n6, C20:3n6, C20:3n3, C20:4n3, C22:0, C22:1n11, C22:1n9, C22:1n7, C21:5n3, C22:4n6, C22:5n6, C22:5n3, and C24:0, found in quantity <1 g/100 g total FAME, were utilized for calculating the fatty acid classes but they are not reported in the table.

**Table 8 animals-13-00715-t008:** Conjugated dienes (mmol hydroperoxides/100 g freeze-dried prawn muscle; CD) and thiobarbituric acid reactive substances (malondialdehyde equivalents on 100 g freeze-dried prawn muscle; TBARS) measured in the muscle of giant freshwater prawn fed the experimental diets over the 60-day trial.

	HM0	HM3	HM20	*p* Value
CD	0.21 ± 0.02	0.23 ± 0.04	0.21 ± 0.04	0.71
TBARS	0.56 ± 0.19 ^b^	0.50 ± 0.10 ^ab^	0.28 ± 0.06 ^a^	0.0321

Values are presented as mean ± SD (*n* = 4). ^a,b^ Different letters indicate statistically significant differences among the experimental groups (*p <* 0.05).

## Data Availability

The data presented in this study are available on request from the corresponding author.

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
