# Peer review of "Growth and Welfare Status of Giant Freshwater Prawn (Macrobrachium rosenbergii) Post-Larvae Reared in Aquaponic Systems and Fed Diets including Enriched Black Soldier Fly (Hermetia illucens) Prepupae Meal"

_animals, 2023, doi:10.3390/ani13040715_

Round 1

Reviewer 1 Report

The manuscript has done a good research regarding the replacement of fish meal with black soldier fly prepupae meal in freshwater prawn diet, and for this purpose, some requirement parameters were examined. The writing of the article has been well done from a literary and scientific point of view, and the continuity of the contents has been maintained. It can be accepted in the Journal by making the following corrections.

1-     The scientific name of the insect should be given in the title and the dot at the end of the title should be removed.

2-     The abstract is very general and does not represent the work done. It is better to summarize the topic in two lines and include some important results.

3-     The article does not have the characteristics of fly mead and should be listed at the beginning of materials and methods.

4-     Why were amino acids not measured? By doing that, the research would be more complete.

5-     In line 195, some information is not provided completely. How many times a day was the feeding done and at what time? Did the biometry during the period with what time’s interval?

6-     The value of C14: 0 in table 7 is wrongly written.

7-     In line 445, the high amount of DHA in the shrimp muscle of the control group may be related to its high amount in the diet and fish meal, which is a good result for the control group, and due to its importance for the consumer, it is necessary to point out this point well and discuss completely.

8-     A general conclusion is not well written and should include all important results.

Author Response

Dear Reviewer,

Thank you for your comments about the present manuscript. Here, the point by point answers.

1-     The scientific name of the insect should be given in the title and the dot at the end of the title should be removed.

The scientific name of the black soldier fly has been added in the title and the dot has been removed

2-     The abstract is very general and does not represent the work done. It is better to summarize the topic in two lines and include some important results.

Thank you for your suggestion. The abstract has been rewritten to emphasize the most relevant results obtained by the present study (maintaining the suggested number of words suggested by the journal guidelines).

3-     The article does not have the characteristics of fly mead and should be listed at the beginning of materials and methods.

We are sorry, but we don’t understand the request. If the Reviewer refers to the fly meal, the features of the growth substrate are reported in the paragraph 2.1. Particularly, black soldier fly larvae were fed with coffee silverskin enriched with 15% of spirulina added with distilled water to reach the specific moisture (~70%). The amount of growth substrate was calculated to provide a feeding rate for each larva of 100 mg/day. New substrate was added once a week. These are all the features that characterized the black soldier fly larval rearing.

4-     Why were amino acids not measured? By doing that, the research would be more complete.

The amino acids profile was not measured due to the high similarity to that of fish meal (data well established by many past papers) and the fact that the amino acids profile is not much influenced by the insect diet. Since the main drawback associated to the use of full-fat HM is related to the fatty acid profile (which can really be modulated by the growth substrate), the present research was mainly focused on this aspect. We are sorry for not being able to provide the amino acid profile.

5-     In line 195, some information is not provided completely. How many times a day was the feeding done and at what time? Did the biometry during the period with what time’s interval?

We have added additional information about the feeding rate in the revised version of the MS (lines 219-223). Particularly, the daily ration (10% body weight) was divided in two equal amounts: one provided in the morning and one in the afternoon. In addition, the daily feeding ration was computed every 2 weeks in intermediate measurement conducted on a representative number of specimens per tank (without sacrifice them).

6-     The value of C14: 0 in table 7 is wrongly written.

The reviewer is right. Actually, the mistake comprises the first three lines of the Table 7 (total lipids, c14 and c16) due to a typographical error. Table 7 has been corrected.

7-     In line 445, the high amount of DHA in the shrimp muscle of the control group may be related to its high amount in the diet and fish meal, which is a good result for the control group, and due to its importance for the consumer, it is necessary to point out this point well and discuss completely.

Thank you for your suggestion. A sentence has been added in the discussion (Lines 477-481) of the revised manuscript).

8-     A general conclusion is not well written and should include all important results.

The conclusion has been partially rewritten including the most relevant results of the present study.

Reviewer 2 Report

Review

Paper title: Growth and welfare status of giant freshwater prawn (Macrobrachium rosenbergii) post larvae reared in aquaponic systems and fed diets including enriched black soldier fly prepupae meal

The authors conducted a classical laboratory study to reveal the effects of two diets based on black soldier fly prepupae meal on post larvae of the giant freshwater prawn Macrobrachium rosenbergii under controlled conditions. The experimental diets were shown to have no effects on the survival rate, growth performance, molting regulation, morphology, health and nutritional status of hepatopancreas after a 60-d period of cultivation. The authors concluded that their prepared diets are appropriate for post larvae of this species.

All these reasons explain the relevance of the paper by Matteo Zarantoniello and co-authors submitted to "Animals".

General scores.

The data presented by the authors are original and significant. The study is correctly designed and the authors used appropriate sampling methods. In general, statistical analyses are performed with good technical standards. The authors conducted careful work that may attract the attention of a wide range of specialists focused on prawn aquaculture.

Recommendations.

L 47-88. Please, separate this section into 2-3 paragraphs.

The authors used ANOVA to test the data for differences. This parametric approach requires normal data distribution and data heterogeneity. Thus, the authors should test the data for normality and heteroscedasticity and transform the data if required or they should use a non-parametric approach.

Specific remarks.

L 29. Consider replacing “interest of” with “interest in”

L 31. Consider replacing “high percentage” with “a high percentage”

L 32. Consider replacing “On this regard” with “In this regard”

L 39. Consider replacing “associated to” with “associated with”

L 57. Consider replacing “been firstly” with “been first”

L 58. Consider replacing “been firstly” with “been first”

L 59. Consider replacing “without negatively affect growth” with “without negative effects on growth”

L 66. Consider replacing “On this regard” with “In this regard”

L 69. Consider replacing “aquafeeds sustainability” with “aquafeed sustainability”

L 71. Consider replacing “as major” with “as a major”

L 77. Consider replacing “that limit” with “that limits”

L 82. Consider replacing “On this regard” with “In this regard”

L 101. Consider replacing “in accord to” with “in accordance wuth”

L 109. Consider replacing “the prawns culture” with “the prawn culture”

L 114. Consider replacing “insects rearing” with “insect rearing”

L 119. Consider replacing “including a” with “including”

L 120. Consider replacing “a final moisture” with “final moisture”

L 125. Consider replacing “adding new” with “adding a new”

L 132. Consider replacing “diets preparation” with “diet preparation”

L 143. Consider replacing “approximatively” with “approximately”

L 145. Consider replacing “and pellet” with “and the pellets”

L 152. Consider replacing “lipids according” with “lipids according to”

L 175. Consider replacing “acclimation period” with “the acclimation period”

L 189. Consider replacing “at constant temperature” with “at a constant temperature”

L 194. Consider replacing “Feeding  trial  duration” with “The feeding  trial  duration”

L 198. Consider replacing “Feed particle size were” with “The feed particle size was”

L 247. Consider replacing “as fluorescent” with “as a fluorescent”

L 266. Consider replacing “consisting in” with “consisting of”

L 269. Consider replacing “profile” with “profiles”

L 301. Consider replacing “from HM20” with “from the HM20”

L 302. Consider replacing “from both HM0” with “from both the HM0”

L 317. Consider replacing “samples form” with “samples from”

L 317. Consider replacing “fed HM3 and HM20 diets compared to those fed on” with “fed the HM3 and HM20 diets compared to those fed on the”

L 356. Consider replacing “in HM3 and HM20 groups” with “in the HM3 and HM20 groups”

L 359. Consider replacing “HM0 group” with “the HM0 group”

L 361. Consider replacing “HM0 and HM3” with “the HM0 and HM3”

L 363. Consider replacing “in HM3” with “in the HM3”

L 365. Consider replacing “from HM0” with “from the HM0”

L 381. Consider replacing “HM3 group” with “the HM3 group”

L 399. Consider replacing “possible alternative” with “a possible alternative”

L 406. Consider replacing “used in” with “used in the”

L 407. Consider replacing “relative high” with “a relatively high”

L 410. Consider replacing “from HM20” with “from the HM20”

L 420. Consider replacing “from HM3 to HM20 diets and in the presence of carotenoids in” with “from the HM3 to HM20 diets and in the presence of carotenoids in the”

L 442. Consider replacing “shorter chain” with “shorter-chain”

L 443. Consider replacing “reflected the one” with “reflected that”

L 471. Consider replacing “a successful molting” with “successful molting”

L 476. Consider replacing “a further” with “further”

L 481. Consider replacing “On this regard” with “In this regard”

L 490. Consider replacing “On this regard” with “In this regard”

L 497. Consider replacing “associated with a” with “associated with”

L 511. Consider replacing “different composition” with “different compositions”

L 520. Consider replacing “crustacean” with “crustaceans”

Author Response

Dear Reviewer,

Thank you for all your specific remarks which have been all applied in the revised version of the manuscript. We also divided the first part of the introduction in 3 paragraphs as suggested. Regarding the statistical approach, the reviewer is right. We selected one-way ANOVA (instead of a non-parametric test) in light of the results obtained by previous statistical test applied to our datasets. Unfortunately, we forgot to include a sentence (now inserted in the revised version of the manuscript) that specifies that all the data were checked for normality and homoscedasticity (and obviously the tests that we used). Thank you for your comment.

Round 2

Reviewer 1 Report

The meaning of characterize of fly meal is how much protein, fat, ash and fatty acid it has. If there is an analysis of fly meal, please add it briefly to the article.

Author Response

Unfortunately, we are not able to provide the amount of protein, fat, ash and fatty acid of the insect growth substrate used for the present study. The reason is that the present study is part of a wider project in which the different steps (from the insect production to the application of insect-based diets to farmed aquatic species) produce a great amount of data that have been divided among the different partners. We handle the aquaculture part, so we reported the insect rearing condition to provide the reader all the information of the experimental set up. The results obtained form analyses on the fly meal are handled by our colleagues and will be probably involved in a further paper more focused on the insects’ growth performances and health status and targeted to a Journal of Entomology.

However, we have the proximate composition (%DM) of the black soldier fly prepupae meal used in both the present and for a previous one (Ratti et al. [33]). We cannot report the same results already published in another paper, but to provide more information we add a mention in the text to Ratti et al. [33] that reports the proximate composition of the same HM used for the present study.

We hope that this information makes the present manuscript more complete.